# The Multi-Hot Representation-Based Language Model to Maintain Morpheme Units

**Ju-Sang Lee, Joon-Choul Shin and Choel-Young Ock ***

Department of Electrical, Electronic and Computer Engineering, University of Ulsan, Ulsan 44610, Korea
* Correspondence: okcy@ulsan.ac.kr

**Abstract:** Natural language models brought rapid developments to Natural Language Processing (NLP) performance following the emergence of large-scale deep learning models. Language models have previously used token units to represent natural language while reducing the proportion of unknown tokens. However, tokenization in language models raises language-specific issues. One of the key issues is that separating words by morphemes may cause distortion to the original meaning; also, it can prove challenging to apply the information surrounding a word, such as its semantic network. We propose a multi-hot representation language model to maintain Korean morpheme units. This method represents a single morpheme as a group of syllable-based tokens for cases where no matching tokens exist. This model has demonstrated similar performance to existing models in various natural language processing applications. The proposed model retains the minimum unit of meaning by maintaining the morpheme units and can easily accommodate the extension of semantic information.

**Keywords:** language model; tokenization; multi-hot representation; maintain morpheme units; morpheme and syllable-base tokens

## 1. Introduction

Natural Language Modeling is a technique that uses the information surrounding a particular position within a set of natural languages, such as sentences or context, to predict the natural language that is likely to appear in that position. Language models are used as a pre-training concept in other natural language processing techniques. Representative models include Embeddings from Language Model (ELMo) [1], Bidirectional Encoder Representations from Transformers (BERT) [2], and Generative Pre–Training (GPT) [3], which are large-scale deep learning models. The development of such language models has brought significant changes in natural language processing research.

Representative language models have used encoder–decoder-based transformer models [4]. Typical language models include BERT and GPT. BERT uses the encoder part of the transformer model to learn text representations, while GPT uses the decoder part. In addition, BERT has an improved understanding of contexts and sentences, while GPT possesses the strength of generating natural languages using the inputted information. Research on BERT has mainly been conducted to improve the model's problems. Some models that have modified the existing BERT's training process to address its problems include RoBERTa (a Robustly optimized BERT pretraining Approach) [5], ALBERT (A Lite BERT) [6], and ELECTRA [7]. Studies have also been conducted on models such as Sense-BERT and KnowBERT that use lexical semantic information such as WordNet [8,9]. Other transformer encoder language models include CharBERT and Charformer [10,11]. Those models use a character-based tokenizer. Conversely, research on GPT has focused on the expansion of the language model's size, leading to the release of GPT-2 and GPT-3 [12,13]. These models expanded the size of the model and training data to increase their capacity to represent larger amounts of data.

Recent studies have presented standards for processing and handling the exponentially increasing amounts of corpus data needed to train a language model. Research has also been conducted on T5 (Text-to-Text Transfer Transformer) [14], a model that predicts sentences based on the inputted text, rather than predicting correct answers as done by existing language models.

Byte Pair Encoding (BPE) is a tokenizer for language models [15]. BPE generates a list of tokens to be conclusively used through the process of combining high-frequency combinations that appear in documents using a data compression algorithm. However, BPE does not reflect the characteristics of the language to be used as it simply performs combinations based on frequency. Using the BPE-based tokenization method poses the following issues:

- Differences in morpheme separation and unknown token ratios due to dependence on the corpus used for tokenization;
- Loss of meaning due to morpheme separation.

In the BPE-based token generation method, a loss of morpheme information occurs in the process of converting morphemes into tokens. For example, the newly coined word 'staycation' (a compound of 'stay' and 'vacation') is separated into 'stay' and 'cation' by BPE. However, if 'staycation' does not appear in the training data or the frequency is low, a problem occurs. Since 'cation' is often used as a chemical term, the meaning of 'vacation' may disappear, leading to a possible loss of meaning of 'staycation'. BPE-based composition methods have posed issues when applying the various lexical semantic information about words. In addition, the separation of morphemes has rendered it difficult to apply the semantic information of vocabularies such as WordNet—an English lexical database of semantic relations [16].

The one-hot representation is the easiest way to express natural language, and by embedding it, many language processing models use it. In the case of one-hot representation, there is a problem in that the embedding table becomes large when the number of types of natural language to be expressed increases. For example, in the case of Chinese, there are simply many characters, and in the case of English or Korean, there are many words that can be generated according to the combination of characters. Therefore, the use of one-hot encoding increases the proportion of the embedding matrix in the overall model. Many studies have been conducted to solve the problem of one-hot expressions. K-way D-dimensional Discrete Embedding (KD encoding) expresses a natural language as a K-way D-dimensional code rather than a one-hot, embedding it and expressing it as a vector [17]. One the other hand, Multi-hot compact network embedding (MCNE) and Effective Multi-hot encoding and classification modUle (EMU) have introduced a method to reduce the size of a model using multi-hot representation [18,19]. These two studies did not use multi-hot expressions for language models, but multi-hot expressions were used to reduce the number of objects to express with embeddings.

We propose a multi-hot representation-based language model to maintain morphemes in token-based language models. We chose Korean as the language to be used in the experiment because Korean consists of several combinations with one character, and different morphemes can be applied to a subword owing to its characteristics as an agglutinative language. Additionally, there is a limit to the number of words that can be expressed in BPE because the number of words created by the combination of letters in Korean is large. The NIKL MODU Corpus: Newspaper Corpus (https://corpus.korean.go.kr, (accessed on 25 August 2020)) was used as training data for the language model, and UTagger, a morpheme analyzer developed by the University of Ulsan, was used to conduct a morpheme analysis on the corpus [20]. In addition, a multi-hot representation method—a set of one-syllable tokens—was used for words not included in the token list. Syllables were tokenized by extracting high-frequency syllables from the corpus, where the proposed model combined these with an existing set of morpheme-based tokens. All tests were conducted using an existing morpheme-based language model as a comparative model. The performance of each model was evaluated in four areas.

## 2. Materials and Methods

### 2.1. Input for the Multi-Hot Language Model

We propose a language model based on multi-hot representation. In existing language models, a specific result value is derived for each token after converting an inputted sentence into tokens. However, this method poses challenges when a single morpheme is represented as a set of syllable tokens, and tokens may exceed the maximum length when the number of generated input tokens increases. However, the existing subword method is used only for verbs.

The multi-hot LM uses a method that generates a single combined token for a set of syllable tokens. Figure 1 shows how the example "Staycation is trending" is generated as input to the language model. If no perfect match exists for 'Staycation', it is converted into a set of syllable tokens, and the token value for that morpheme is generated by multiplying the embedding information for each syllable by the position embedding within the set of tokens.

$$O_t = \sum_i T_i T P_i \tag{1}$$

The set of syllable tokens is converted into a single token using the above equation. $T_i$ is the token embedding value, and $TP_i$ is the token position embedding value within the syllable token set. By multiplying the token position embedding value with the token embedding value, the syllable tokens with the same combinations but different syllable positions can be differentiated.

$$Input_i = O_i + Pos_i + P_i \tag{2}$$

The above equation represents the process in which the input to the transformer encoder is conclusively generated. $Pos_i$ is the part-of-speech embedding value for the corresponding token, and $P_i$ represents the sentence position embedding, which has also been used in existing models.

| Language model Input | | $Input_0$ | | | | | | | | | $Input_1$ | $Input_2$ | |
|---|---|---|---|---|---|---|---|---|---|---|---|---|---|
| Create Input | Pos Embedding | NN | | | | | | | | | VBZ | VBG | |
| | Sentence Position Embedding | 0 | | | | | | | | | 1 | 2 | |
| | Token Value | $T_0$ | | | | | | | | | $T_1$ | $T_2$ | |
| Multi-hot Tokenizer | Token | S | t | a | y | c | a | t | i | o | n | is | trend | ing |
| | Token Position | 0 | 1 | 2 | 3 | 4 | 5 | 6 | 7 | 8 | 9 | 0 | 0 | 1 |
| morphology parsing | morpheme | Staycation(Noun: NN) | | | | | | | | | is(Verb: VBZ) | trending(Verb: VBG) | |

**Figure 1.** The multi-hot representation language model acting on an input value (on the example: "Staycation is trending").

### 2.2. Loss Function for the Multi-Hot Language Model

The multi-hot representation language model proposed in this paper uses BERT's training method. The multi-hot LM uses the same masked LM process as BERT, but more than one correct token can exist. Because the multi-hot LM can have more than one answer, the existing loss function was modified. In addition, the final output value was obtained by adding one linear layer without applying the embedding table used for the input to the final layer.

The existing masked LM uses Softmax cross entropy (SCE), which is a function that is used for problems with a single correct answer. However, this function is difficult to use in the multi-hot LM proposed in this paper.

$$SCE = -\sum_i \log(S(O_i)) y_i \tag{3}$$

The multi-hot LM can have several correct answers for one masking position. When multiple answers exist, the $y_i$ value increase with the number of correct answers. Additionally, since the model is trained so that the probability of the correct answer is 1, when the position of a correct answer increases, the loss value of another true position increases and does not fall below a specific loss value when multiple answers exist.

Another loss function is binary cross entropy (BCE), which finds a loss value for multiple correct answers. The average of the loss value for each output position is used.

$$BCE = -\frac{1}{N}\sum_i^N \updownarrow_i, \ \updownarrow_i = y_i(\log(p(y_i)) + (1 - y_i)\log(1 - p(y_i))) \tag{4}$$

This poses a problem for masked LM, as outputs are produced according to the list of tokens, and, with most answers being zero, the model is trained to elicit a probability value of zero for each output.

For the multi-hot LM, we implemented a function to obtain loss values for problems with more than one answer by modifying the SCE. We changed the probability of a correct answer from 1 to $y_i/n$. Here, n is the total number of correct answers, and $y_i$ is the same number of tokens included in the correct answer. By adjusting the probability value of the correct answer, the sum of the Softmax function is designed to not exceed 1 and to prevent excessive loss values. For example, for the masked word 'Archimedes' (a person's name, with multi-hot token representation: ['@A', '@r', '@c', '@h', '@i', '@m', '@e', '@d', '@e', '@s']), the resulting probability value for '@A' is $1/10$, and the total probability value for all ten letters is 1. The '@' symbol was used to distinguish it from tokens created via the existing BPE. The problem with using Softmax is that the probability value of the correct answer of the tokens at each position becomes 1, resulting in a total of 10. In addition, if one of the three answers' loss decreases, the other answer increases the loss value. Therefore, the model cannot be trained to converge. We calculate a loss value for $O_i$, which is the output value $\updownarrow_i$:

$$\updownarrow_i = -(\log(S(O_i)) - \log(y_i/n)) \tag{5}$$

Here, $S$ is the Softmax function. The above equation is the existing SCE, minus $\log(y_i/n)$. Since the multi-hot representation-based loss function gives an answer probability of $y_i/n$, we infer that the log-Softmax value of the output value at that position converges to zero using $\log(y_i/n)$. For duplicate characters, $y_i$ increases, so the probability of that character increases with the number of characters. For example, for a set of three different tokens, the maximum value of $\log(S(O_i))$ cannot be less than $\log(1/3)$ as the probability of each answer position is $1/3$. So, $\log(1/3)$ is subtracted to derive a loss value such that the existing probability value is 1. After obtaining $\updownarrow_i$—the loss value for each output neuron—the correct answer for the current training data is redefined using this value. This process is used to prepare for the case where the probability value of the model exceeds $y_i/n$.

$$label_i = \text{if } \updownarrow_i < 0 : y_i \ else \ : 0 \tag{6}$$

We redefine the answer value in the training data to consider cases where the loss value of the answer position exceeds the maximum value in a multi-hot LM. When the loss value of each output neuron becomes positive, it results in a loss value that exceeds the target probability. The average loss value decreases if a positive $\updownarrow_i$ value is used as the loss value for the current training data, and the output value at the respective position becomes less likely to be correct as the training progresses. The correct answer was redefined to exclude positive $\updownarrow_i$ values. When $\updownarrow_i$ is positive, the result value label is 1, and in other cases, it is 0. The $\updownarrow_i$ value—the modified loss value for multiple answers—and $label_i$—the redefined answer—were conclusively used to find the loss value for the current training data.

$$\Delta Loss = \sum_i(\updownarrow_i label_i) / \sum_i(label_i) \tag{7}$$

We then find the final loss value for masked embedding that can have multiple correct answers used as training data in a multi-hot LM. The average of the redefined answer value $label_i$ multiplied by the loss value $\updownarrow_i$ was used as the final loss value. The average value used as the multi-hot LM has more than one correct answer, and simple summation results in a large loss value being applied to the model, as in the existing loss function.

### 2.3. Configuration of the Model and Token List

For all models, we used the 'NIKL MODU Corpus: Newspaper Corpus' (16GB) as the training data. Masked LM and Next Sentence Labeling were used for training, as in the existing BERT model. We used models of two sizes for this test. The size settings are described below; all other settings apart from size remained the same.

- Small Model—4 transformer layers, 256-neuron hidden layers, maximum token input size of 256 tokens;
- Basic Model—12 transformer layers, 768-neuron hidden layers, maximum token input size of 512 tokens.

A test was conducted with different token configurations to evaluate the multi-hot LM. Three methods were used: (1) the existing BPR method, (2) a combination of BPE and syllable tokens, and (3) a combination of a dictionary-based token list and syllable tokens.

Table 1 shows the number of tokens used for each model. The list that combined syllable tokens with the existing BPE base was used to evaluate the model's performance when syllable tokens were added to the existing model. This model was constructed by removing English alphabets, numerals, and Chinese characters from the list of tokens generated from the existing BPE base, removing tokens combined with parts-of-speech from the morpheme-based BPE, and eliminating duplicates. The existing BPE base consists of a token list with morphemes alone or morphemes combined with parts-of-speech.

**Table 1.** Number of tokens by generation method.

| Model | BPE Base | BPE + Syllable | Dictionary + Syllable |
|---|---|---|---|
| All Tokens | 37,027 | 20,892 | 37,822 |
| Syllable Tokens | · | 1994 | 1994 |

The dictionary-based token list was used to evaluate the performance of the multi-hot LM for token lists using high-frequency dictionary words. This list extracted high-frequency nouns and predicates from the words listed in the dictionary, and adverbs were included. Compound nouns were excluded from nouns, and syllable tokens were finally added to the list.

### 3. Results

#### 3.1. Details of the Experimental Areas

We tested and verified the performance of the multi-hot LM in four evaluation areas. These were KorQuAD v1.0 (https://korquad.github.io/KorQuad%201.0, (accessed on 28 January 2019)), Named Entity Recognition (NER), Semantic Role Labeling (SRL), and Naver Sentiment Movie Corpus (NSMC) (https://github.com/e9t/nsmc, (accessed on 28 June 2016)). All the data used a morpheme-analyzed corpus using UTagger, and answers in raw corpus format were reprocessed according to morpheme units. The following is a detailed description of the four areas.

- KorQuAD v1.0—A Korean QA corpus constructed by LG CNS. It consists of 60,407 training data and 5774 validation data. This corpus was used to compare machine reading comprehension performance with that of existing language models. An exact match for the answers and the F1-score were used for the evaluation.
- NER and SRL (NIKL MODU Corpus)—The 'NIKL MODU corpus', constructed by the National Institute of Korean Language, was used for NER and SRL. The two domains

were used to test the model's semantic analysis of natural languages. NER consists of 15 tags, while SRL consists of 18 tags. A BIO-based method was used for NER, and for SRL, the word segments of the target predicates were attached after the natural language sentence to be analyzed. The NER corpus contains 120,066 sentences of training data and 30,018 sentences of test data. The SRL corpus contains 108,509 sentences of training data and 27,501 sentences of test data. The F1-score was used for the evaluation.

- NSMC (Naver Sentiment Movie Corpus)—The NSMC analyzes the positive and negative emotions in review comments. Unlike the above domains, NSMC was used to determine the suitability of the language model for a spoken corpus in which everyday terms were used, rather than a written corpus like the newspaper corpus. The corpus contains 150,000 training data and 50,000 test data. It was evaluated based on the accuracy.

Figure 2 represents the model used in each experimental area. In order to compare the performance of the language models, the experimental model was simply constructed. This was to reduce the influence on the performance of the experimental area by other factors.

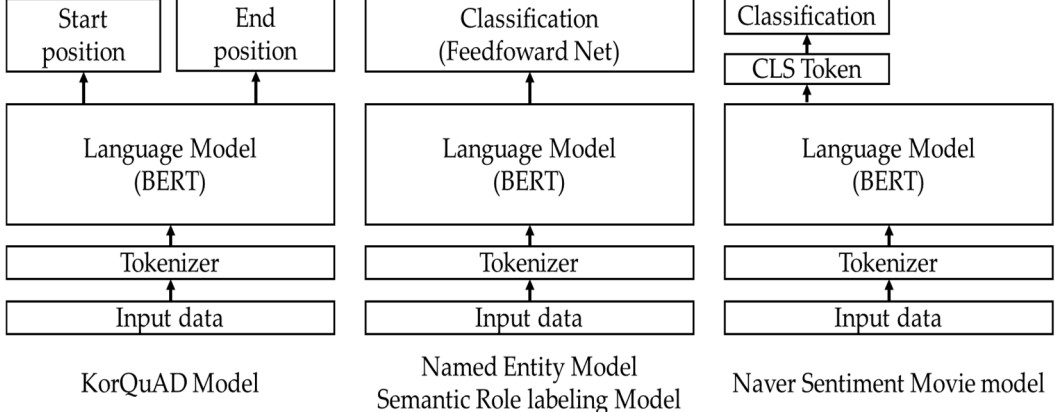

**Figure 2.** Experimental area (KorQuAD v1.0, NER, SRL, NSMC) usage model.

*3.2. Comparison of Loss Functions*

We evaluated the loss functions for the multi-hot LM. Each model used a BPE-based syllable unit token list, which was applied to the small model. Under the same training conditions, the loss function was structured and operated in three ways in masked embedding. These included the existing SCE, BCE, and a variant of SCE for the multi-hot representation proposed in this paper. The following are the test results on KorQuAD v1.0.

In Table 2, for multi-hot LM, the training method using the proposed multi-hot-based SCE was the most suitable. The existing SCE and BCE methods are unsuitable for language models with multiple correct answers.

**Table 2.** Results for KorQuAD v1.0 regarding the loss function; list of tokens based on BPE and syllables, tested with a small model.

| Loss Function | Exact | F1-Score |
|---|---|---|
| Softmax Cross Entropy | 69.8% | 76.3% |
| Binary Cross Entropy | 43.3% | 55.8% |
| Multi-hot Softmax Cross Entropy | **73.6%** | **80.1%** |

*3.3. Experiments on the Multi-Hot Language Model*

The performance of the multi-hot LM that maintained the morpheme units proposed in this paper was evaluated with regard to natural language processing and compared with the existing models. The training and evaluation of the test domains were conducted

within the same hardware environment. The same training settings were used for testing models of the same size.

In Table 3, the evaluation results of the model proposed in this paper and the existing models for each domain are presented by token configuration and model size. Among the four domains, KorQuAD v1.0—which was used to test machine reading comprehension—showed lower performance than the existing BPE base. The remaining three domains showed higher performance, excluding SRL in the small models. A detailed analysis of each domain was conducted to determine the cause of the differences from the existing model. First, a detailed evaluation of the correct answers was conducted on the test results from KorQuAD v1.0.

**Table 3.** Model test for KorQuAD v1.0, Named Entity Recognition, Semantic Role Labeling, and the Naver Sentiment Movie Corpus; scores in bold are the best overall.

| Area | | KorQuAD v1.0 | | NER | | SRL | | NSMC |
|---|---|---|---|---|---|---|---|---|
| | | | | **F1-Score** | | **F1-Score** | | |
| **Token List** | **Size** | **Exact** | **F1-Score** | **Macro** | **Micro** | **Macro** | **Micro** | **Accuracy** |
| BPE base | Small | **75.2%** | 81.2% | 78.65% | 86.78% | **43.41%** | **54.35%** | 86.99% |
| BPE + Syllable | Small | 73.6% | 80.1% | **79.81%** | **87.39%** | 43.18% | 53.21% | 87.62% |
| Dictionary + Syllable | Small | 74.9% | **81.5%** | 79.70% | 87.37% | 43.07% | 53.20% | **87.97%** |
| BPE base | Basic | **83.0%** | **89.1%** | 80.74% | 88.57% | 45.79% | 57.12% | 87.95% |
| BPE + Syllable | Basic | 81.8% | 87.9% | **81.47%** | **89.01%** | **46.54%** | **58.64%** | **88.91%** |

Subsequently, an analysis of the performance differences with existing models was conducted. In addition, the results were divided and analyzed in relation to the exact match ratio according to the answer form. The test consisted of the case where at least one morpheme was separated and the case where no morphemes were separated when the morpheme of the correct answer was tokenized. The analysis results were extracted from the small models.

In Table 4, the comparison showed a large number of correct answers for unseparated morphemes in the existing BPE model but lower performance than the dictionary-based syllable token combinations when separation occurred. The table shows that maintaining morphological forms has a significant effect on performance.

**Table 4.** Ratio of exact matches based on the separation of morphemes of KorQuAD v1.0 in the small model.

| Token List | KorQuAD Exact Match Ratio | | | |
|---|---|---|---|---|
| | **Not Divided** | | **Divided** | |
| BEP base | **2046/2618** | **78.15%** | 2301/3156 | 72.90% |
| BPE + Syllable | 1958/2618 | 74.78% | 2291/3156 | 72.59% |
| BPE + Dictionary | 2002/2618 | 76.47% | **2326/3156** | **73.70%** |

The small models performed well in NER but not in SRL. However, the multi-hot LM showed high performance in both domains for the basic models. Proper nouns, which frequently appear as answers in NER, appear less frequently in corpora and show more diverse combinations than does commonly used vocabulary. Therefore, the multi-hot LM showed high performance as morphemes are often separated or cannot be tokenized in the existing BPE base. The following table shows the F1-score results of 3 main tags from a total of 15 tags in the NER for each model. The selected tags were person, location, and organization. These are typical categories of named entities that are in the form of proper nouns. The performance comparison for these tags was based on three small models.

Table 5 shows higher performance for the set of named entities with a high number of proper nouns in the model that is represented as a set of syllables than in the existing model.

The set of syllables used in the multi-hot LM has a positive effect on performance for proper nouns. For SRL, the existing model and the multi-hot LM showed good performance in the small and basic models, respectively. The diversity of the multi-hot representation increased with the number of parameters used in the model, showing better performance than the existing model.

**Table 5.** Details of named entity recognition results in small language models; PER (person), LOC (location), and ORG (organization).

| Token List | NER Tag | | |
|:---:|:---:|:---:|:---:|
| | PER | LOC | ORG |
| BEP base | 93.84% | 87.53% | 84.26% |
| BPE + Syllable | **94.95%** | 87.68% | **85.18%** |
| BPE + Dictionary | 94.77% | **87.81%** | 85.09% |

Finally, the multi-hot LM performed better in all areas than the existing models in the NSMC. The incorrect separation of morpheme units affected the results in areas with highly diversified and frequent informal expressions, such as dialog. In this regard, the multi-hot LM is effective.

## 4. Conclusions

Herein, we proposed a multi-hot representation-based language model to maintain morpheme units in morpheme-based language models. To represent and train a multi-hot representation method, we proposed generating combined tokens with a set of syllable tokens to be used as input to a transformer model. To train these combined tokens, the language model was trained using a loss function that can apply changing answer probabilities according to the number of correct answers, rather than using the existing loss function. The model was evaluated in several natural language processing domains to test its performance. Each domain was tested by changing the size of the model and token list. The model exhibited better performance than the existing model in all areas, excluding machine reading comprehension.

The multi-hot representation-based language model has several advantages because of the retention of morpheme units. Maintaining the minimum unit of word forms aids in restricting the result values of the language model to morphemes. In addition, these results can be easily applied to the semantic knowledge of natural languages if they are output as minimum units of word forms. The multi-hot LM can set the foundations for research on methods to use semantic information with current language models. Finally, although multi-hot representation was applied only to Korean in this paper, it can be easily applied to other natural languages. It is easily accessible because the set of characters of the corresponding natural language is simply added to the existing token list, and only the location information of each token is additionally used. Further, it is a model that does not have any special dependencies on a specific natural language.

**Author Contributions:** Data curation, J.-S.L. and J.-C.S.; formal analysis, J.-S.L. and J.-C.S.; methodology, J.-S.L.; project administration C.-Y.O.; writing—original draft, J.-S.L.; writing—review and editing, J.-S.L., J.-C.S. and C.-Y.O. All authors have read and agreed to the published version of the manuscript.

**Funding:** This research received no external funding.

**Institutional Review Board Statement:** Not applicable.

**Informed Consent Statement:** Not applicable.

**Acknowledgments:** This work was supported by an Institute for Information & Communication Technology Promotion (IITP) grant funded by the Korea government (MISP) (No. 2013-0-00131 Development of Knowledge Evolutionary WiseQA Platform Technology for Human Knowledge Aug-

mented Services) and the Basic Science Research Program through the National Research Foundation of Korea (NRF) funded by the Ministry of Education (No. 2020R111A3070938).

**Conflicts of Interest:** The authors declare no conflict of interest.

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
