# Peer review of "The Multi-Hot Representation-Based Language Model to Maintain Morpheme Units"

_applsci, doi:10.3390/app122010612_

Round 1

Reviewer 1 Report

This paper mainly proposed a multi-hot language representation model for Korean language and tested via different corpus such as NER and sentiment analysis.  Two major disadvantages of this paper are that:

1. Multi-hot language representation is not new. The proposed model did not cite previous works and in fact no articles related to multi-hot representation is reported. This does merit for a new contribution in the paper. 

2. Testing the multi-hot language model on different corpus does not reflect its true performance. Perhaps the experiments should focus on one specific domain such as NER or sentiment analysis and compare the effect of the proposed multi-hot representation model against others such as syntactic/semantic-based language representation. This is crucial to prove that the proposed model gives impact to a specific research domain. 

Finally, please observe spacing before punctuations or in-para citations.

Author Response

Dear, reviwer

Thanks for your review. 

Answer 1.

I've found some papers using multi-hot encodings, and I plan to add them.
 1. Multi-Hot Compact Network Embedding
 2. EMU: Effective Multi-Hot Encoding Net for Lightweight Scene Text Recognition With a Large Character Set

Answer 2.

My intention was to experiment with the performance of the multi-hot language model in the existing one-hot language model and various natural language processing domains. So I chose four different areas, and I didn't focus on any particular area. Each area is divided into Machine Reading Comprehension, semantic analysis of sentences, and sentence comprehension for colloquial language (the model learns only written language), and various areas were selected. Also, in the case of a language model, since there are various variables depending on the training data and the model, a comparative experiment was conducted on the model learned in the same environment.

I will check the punctuation of the thesis to be edited or the spacing before citations in paragraphs again.

Thank you for consideration.

Reviewer 2 Report

The paper proposes a language model based on a multi-hot representation of syllables in Korean, with the objective of maintain appropriate morpheme units in morpheme-based language models.

Results give confirm of the effectiveness of the proposed approach.

The only questionable issue is related to the possible extension of such an approach to other languages. Since this is an aspect the authors do not take into consideration, it would be useful to add a couple of lines on this: how can the proposed model be useful for other languages?

Some minor detailed comments follow:

- line 60: I suggest you to remove "occurring"

- line 98: pay attention to the citing function (maybe the ref is missing in your bib file)

- line 186: Table 2 should be Table 1 (wrong label)

Author Response

Dear, reviewer

Thanks for your review.

I've added the possibilities for other languages to the conclusion of Chapter 4.  And corrected for reference errors and wording.

Thank you for consideration.

Round 2

Reviewer 1 Report

Please observe minor formatting issues as in document attached. Also, please cite the first time Multi-Hot representation model is mentioned in the paper.

Author Response

Dear, reviewer.

I modified the format of the thesis by referring to the template.

I added it as a reference, thinking that the K-way D-dimentional encoding [17] paper published in 2018 was the first proposed for multi-hot encoding.

Thank you for consideration.